# Cancer Cell Inhibiting Sea Cucumber (*Holothuria leucospilota*) Protein as a Novel Anti-Cancer Drug

**DOI:** 10.3390/nu14040786

**Published:** 2022-02-13

**Authors:** Ruizhen Ru, Yanzheng Guo, Juanxuan Mao, Zonghe Yu, Wen Huang, Xudong Cao, Huijian Hu, Minjie Meng, Lihong Yuan

**Affiliations:** 1Guangdong Province Key Laboratory for Biotechnology Drug Candidates, School of Biosciences and Biopharmaceutics, Guangdong Pharmaceutical University, Guangzhou 510006, China; ruruizhen@163.com (R.R.); 15917378614@163.com (Y.G.); maojuanxuan@163.com (J.M.); 2College of Marine Sciences, South China Agricultural University, Guangzhou 510640, China; yuzh@scau.edu.cn; 3Fisheries Research Centre, Key Laboratory of Animal Nutrition and Feed Science in South China of Ministry of Agriculture and Rural Affairs, Guangdong Key Laboratory of Animal Breeding and Nutrition, Institute of Animal Science, Guangdong Academy of Agricultural Sciences, Guangzhou 510640, China; huangwen549@126.com; 4Department of Chemical and Biological Engineering, University of Ottawa, Ottawa, ON 999040, Canada; xcao@uottawa.ca; 5Institute of Zoology, Guangdong Academy of Sciences, Guangzhou 510260, China; 13570909977@139.com

**Keywords:** sea cucumber protein, anti-cancer activity, targeted effects, cell apoptosis, cell migration, marine extractions

## Abstract

Cancer remains the primary cause of death worldwide. To develop less toxic anti-cancer drugs to relieve the suffering and improve the survival of cancer patients is the major focus in the anti-cancer field. To this end, marine creatures are being extensively studied for their anti-cancer effects, since extracts from at least 10% of the marine organisms have been shown to possess anti-tumor activities. As a classic Chinese traditional medicine, sea cucumbers and compounds extracted from the sea cucumbers, such as polysaccharides and saponins, have recently been shown to exhibit anti-cancer, anti-inflammatory, and anti-oxidant effects. *Holothuria leucospilota* (*H. leucospilota*) is a tropical edible sea cucumber species that has been successfully cultivated and farmed in large scales, providing a readily available source of raw materials to support the development of novel marine anti-cancer drugs. However, very few studies have so far been performed on the biological activities of *H. leucospilota*. In this study, we first investigated the anti-cancer effect of *H. leucospilota* protein on three cancer cell lines (i.e., HepG2, A549, Panc02) and three normal cell lines (NIH-3T3, HaCaT, 16HBE). Our data showed that *H. leucospilota* protein decreased the cell viabilities of HepG2, A549, HaCaT, 16HBE in a concentration-dependent manner, while Panc02 and NIH-3T3 in a time- and concentration-dependent manner. We also found that the inhibitory effect of *H. leucospilota* protein (≥10 μg/mL) on cell viability is near or even superior to EPI, a clinical chemotherapeutic agent. In addition, our data also demonstrated that *H. leucospilota* protein significantly affected the cell cycle and induced apoptosis in the three cancer cell lines investigated; in comparison, it showed no effects on the normal cell lines (i.e., NIH-3T3, HaCaT and 16HBE). Finally, our results also showed that *H. leucospilota* protein exhibited the excellent performance in inhibiting cell immigrations. In conclusion, *H. leucospilota* protein targeted the cancer cell cycles and induced cancer cell apoptosis; its superiority to inhibit cancer cell migration compared with EPI, shows the potential as a promising anti-cancer drug.

## 1. Introduction

As the primary cause of death, cancer is responsible for approximate 10 million deaths per year worldwide; the morbidity is increasing in both developing and developed countries [1]. According to the World Health Organization (WHO), lung cancer (1.8 million deaths), colon and rectal cancer (935,000 deaths), liver cancer (830,000 deaths), stomach cancer (769,000 deaths) and breast cancer (685,000 deaths) are the top five cancers that cause the most deaths per year [2]. While significant progress has been made in cancer treatments, such as targeted therapy, cancer immunotherapy, radiation therapy and chemotherapy, these treatments are still inadequate due to either their high costs, questionable efficacies, serious side effects or drug resistance, limiting their wide applications in clinics [3,4,5,6]. Therefore, there is a pressing need to develop novel anti-cancer drugs that are broad-spectrum, high potency, low toxicity and more cost effective.

Marine creatures are an enriched ore of drug development; at least 10% of the marine organism extracts possess anti-tumor properties [7,8]. Marine organism extracts are characterized by high efficiency, safety, lower toxicity and reduced drug resistance [9]. At present, some of the marine organism extracts have demonstrated anti-cancer activity in various preclinical studies; several marine organism-derived drugs, such as the cyclic depsipeptide plitidepsin (Aplidin^®^), Tetrodotoxin (Tectin^®^), and Ralaniten acetate (EPI-506) are already in clinical trials [10]. However, limited marine natural resources and complexities associated with active chemical ingredients constitute bottlenecks in marine drugs development [10,11].

Sea cucumbers (or holothurians), a classic traditional Chinese medicine, have been used in traditional medicine practices for thousands of years in China and many other countries in Asia. They have an impressive character of valuable nutrients, including triterpene glycosides (saponins) [12,13,14,15,16], chondroitin sulfates, glycosaminoglycan [17] and peptides [18] that are recognized as the origin of unique biological and pharmacological functions of sea cucumbers, including anti-angiogenesis [19], anti-coagulation [20], anti-bacterial [21], anti-inflammatory [18,22,23], anti-diabetic [24], and anti-cancer [12,15,16,19,25,26]. Among the approximately 21 species of edible sea cucumber in China, with the exception of the *Apostichopus japonicus*, all are tropical species. With the development of advanced breeding technology of several tropical edible sea cucumbers (such as *Holothuria leucospilota* (*H. leucospilota*) and *Stichopus japonicus*), large-scale breeding and farming of these tropical edible sea cucumbers are possible. As a widespread tropical edible sea cucumber, *H. leucospilota* protein makes up more than 70% of its body wall, which is the main source of bioactive peptides [27]. However, the anti-cancer effect of protein generated from *H. leucospilota* remains unexplored. In this study, we aim to explore a possible anti-cancer effect of the *H. leucospilota* protein by investigating the effects of *H. leucospilota* protein on cancer cell survival, migration, cell cycle, and apoptosis.

## 2. Materials and Methods

### 2.1. Reagents

Minute^TM^ Animal Cell/tissue total protein extraction kit (Invent Biotechnologies, Eden Prairie, Minnesota, USA), Cell Counting Kit-8 (CCK-8, US EVERBRIGHT^®^INC., Suzhou, Jiangsu, China), Hoechst-33258 Staining Kit, Annexin V-FITC Apoptosis Detection Kit, Cell Cycle and Apoptosis Analysis Kit (Beyotime, Zhen Jiang, Jiangsu, China), BCA Protein Assay Kit (Keygen Biotech, Nanjing, Jiangsu, China), DMEM medium, fetal bovine serum (FBS), Phosphate Buffer solution (PBS), RPMI-1640 medium (Gibco, Thermo Fisher Scientific, Waltham, MA, USA), penicillin (100 Units/mL), streptomycin (100 μg/mL), and 0.25% Trypsin (1×) (Gibco, Grand Island, NY, United States of America (USA)) were used in this study. Chemotherapeutic agent Epirubicin hydrochloride (EPI), a commonly used anti-tumor drug to treat solid tumors, such as lung cancer, liver cancer, breast cancer, and bladder cancer, was purchased from Aladdin.

### 2.2. Sample Collection and Preparation

Sea cucumber (*Holothuria leucospilota*) were captured from Dapeng’ao Bay of China (Latitude: 22°32′6.95″ N, Longitude: 114°29′12.07″ E). The body wall of sea cucumbers was collected and stored at −80 °C. Total protein was extracted by Minute^TM^ Animal Cell/Tissue total protein extraction kit and quantified by a BCA Protein Assay Kit by following protocols by the vendors. Total protein of *H. leucospilota* was stored at −80 °C until future use.

### 2.3. Cell Lines and Cell Cultures

Three cancer cell lines (Human hepatoma cell HepG2, ATCC^®^ No. HB-8065^TM^; Human lung carcinoma cell A549, ATCC^®^ No. CCL-185^TM^; Human pancreatic cancer cell Panc02, ATCC^®^ No. CRL-2553^TM^), and three normal cell lines (mouse embryonic fibroblasts cell NIH-3T3, ATCC^®^ No. CRL-1658^TM^; Human bronchus epithelial cell 16HBE, ATCC^®^ No. CRL-2078^TM^; Human keratinocyte cell HaCaT was generously provided by the College of Traditional Chinese Medicine of Guangdong Pharmaceutical University as a gift) were used in this study. HepG2, Panc02, NIH-3T3, HaCaT and 16HBE cells were grown in regular growth medium (DMEM) supplemented with 10% fetal bovine serum (FBS), 100 Units/mL of penicillin and 100 μg/mL of streptomycin. In addition, A549 cells were maintained in RPMI-1640 medium supplemented with 10% fetal bovine serum (FBS), 100 Units/mL of penicillin and 100 μg/mL of streptomycin. All cells were incubated at 37 °C with 5% CO_2_ atmosphere.

### 2.4. Cellular Viability Assay

HepG2, Panc02, NIH-3T3, HaCaT and 16HBE cells were seeded on the wells of 96-well plates at a density of 5000 cells per well, and A549 cells were seeded at a density of 4000 cells per well; the cells were incubated overnight. Subsequently, the cells were treated for 24, 48, and 72 h with increasing concentrations of *H. leucospilota* protein in triplicate, after which the Cell Counting Kit-8 was used to determine cell viabilities using a Bio-Rad microplate reader at 450 nm (Hercules, CA, USA) according to the vendor’s protocol. In parallel, EPI (10 μΜ) and PBS (pH 7.4) were also used as positive and blank controls, respectively. According to the protocol, cells were washed twice using PBS, then 10% CCK-8 reagent was added into wells. Three hours later, plates were detected at 450 nm by a microplate reader (Bio-Rad, Hercules, CA, USA). Data were presented as proportional viability (%) by comparing the treatment groups (both *H. leucospilota* protein and EPI treatment groups) with the PBS (pH 7.4) group (blank control), for which the viability was assumed to be 100%. Half maximal inhibitory concentration (IC_50_) values of *H. leucospilota* protein in HepG2, Panc02, NIH-3T3, HaCaT, 16HBE, and A549 cell lines were determined and used for further assay.

### 2.5. Cell Cycle Analysis

The cell cycle distribution was analyzed by flow cytometry using propidium iodide (PI) staining, according to the manufacturer’s instructions. Specifically, HepG2, A549, Panc02, NIH-3T3, HaCaT and 16HBE cells were separately seeded into the wells of 6-well plates at a density of 2 × 10^5^ cells per well and incubated for 24 h, after which the cells were treated with either specific IC_50_ concentrations of *H. leucospilota* protein or EPI (10 μΜ) for 48 h. Subsequently, the cells were trypsinized, collected, and, finally, fixed with 75% EtOH/H2O (*v/v*) at 4 °C overnight. The fixed cells were stained in PI staining solution containing RNase A (1×) for 30 min at 37 °C in the dark. The fluorescent signals were detected by BD FACS Calibur (BD Bioscience). The results were analyzed using FlowJo software (Version 10.0.7, Tree Star, Inc., Ashland, OR, USA).

### 2.6. Cell Apoptosis and Morphology Assay

Apoptotic cells were examined by flow cytometry analysis with an Annexin V-FITC and propidium iodide (PI) double staining. In brief, HepG2, A549, Panc02, NIH-3T3, HaCaT and 16HBE cells were incubated overnight in wells of 6-well plates (2 × 10^5^ cells/well), to which IC_50_ concentrations of *H. leucospilota* protein were added and incubated with the cells for an additional 48 h. Subsequently, the cells were detached, harvested, and incubated for 20 min with 500 μL Annexin V-FITC binding buffer, 5 μL Annexin V-FITC, and 10 μL PI in the dark. The fluorescent signals were detected by BD FACS Calibur (BD Bioscience).

The morphological changes of the nuclei were observed by the staining of the nuclei of the cells with Hoechst-33258. In brief, HepG2, A549, Panc02, NIH-3T3, HaCaT and 16HBE cells were grown on sterile cover slips in 6-well plates treated with the IC_50_ concentrations of *H. leucospilota* protein for 48 h. The cells were fixed in fixative for 15 min, washed twice using PBS (pH 7.4), stained with 500 μL of Hoechst-33258 for 5 min and washed twice with PBS (pH 7.4), mounted with prolonged gold anti-fade reagent, and observed using a fluorescence microscope (Olympus Co, Tokyo, Japan) under 350 nm excitation and 460 nm emission.

### 2.7. Cell Migration Assay

HepG2, A549, Panc02, NIH-3T3, HaCaT and 16HBE cells were grown in 6-well tissue culture plates until confluence. A scrape was carefully made through the confluent cell monolayer with a plastic micropipette tip, washing cellular debris twice with PBS (pH 7.4). Afterward, the IC_50_ concentrations of *H. leucospilota* proteinwere prepared to contain 1% FBS, then were added to cells for 24 h. At the bottom side of each dish, two arbitrary places were marked where the width of the wounds was photographed with an inverted microscope (objective × 4) (Olympus Co, Tokyo, Japan) at time 0 and the 12 and 24 h period time. Finally, the scratch areas were measured by Image J software (National Institutes of Health, Bethesda, MD, USA) and the closure of the wounded area was calculated.

### 2.8. Statistical Analysis

All experiments were performed in triplicates and repeated at least three times. Data are expressed as the mean ± SD. Statistical analysis was performed by unpaired Student’s *t*-test and one-way ANOVA; a *p* < 0.05 was deemed significant.

## 3. Results

### 3.1. The Inhibitory Effects on Cell Viability

As shown in Figure 1, *H. leucospilota* protein reduced the cell viability of HepG2, A549, HaCaT and 16HBE in a concentration-dependent manner, and in atime- and concentration-dependent manner on Panc02 and NIH-3T3. For example, for HepG2 cells, at all concentrations studied (i.e., 0.005–10 μg/mL), *H. leucospilota* protein showed significant cytotoxicity at 24 h; however, at 48 and 72 h, it showed cytotoxicity similar to EPI only at a high concentration of 10 μg/mL. For the case of A549 cells, *H. leucospilota* protein did not show significant cytotoxicity to the cells until the *H. leucospilota* protein concentration reached 10 μg/mL (*p* < 0.001). In contrast, for the case of Panc02 cells, *H. leucospilota* protein demonstrated significant cytotoxicity even at lower concentrations (i.e., 0.005–1 μg/mL) after 48 and 72 h (*p* < 0.001). In addition, its cytotoxicity was much more pronounced at concentrations above 20 μg/mL at 24, 48, and 72 h. For NIH-3T3 cells, *H. leucospilota* protein decreased cell viability and reached a significant level above 5 μg/mL in both 48 and 72 h. In HaCaT cells, *H. leucospilota protein* at high concentration (>5 μg/mL) significantly inhibited the proliferation. For 16HBE cells, *H. leucospilota* protein significantly inhibited cell viability, and was sensitive to 16HBE cells in above 0.1 μg/mL at 24 and 48 h. The IC_50_ concentrations of *H. leucospilota* protein on six cell lines were determined and used for the further assay, as 17.175 μg/mL in HepG2, 10.658 μg/mL (A549), 17.481 μg/mL (Panc02), 8.493 μg/mL (NIH-3T3), 9.282 μg/mL (HaCaT), and 3.489 μg/mL (16HBE), respectively.

### 3.2. Effects of H. leucospilota Protein Treatment on the Distribution of Cell Cycle

It is evident from Figure 2 that, in comparison with EPI, *H. leucospilota* protein caused various effects on the cell cycles of cell lines investigated, suggesting different action mechanisms between *H. leucospilota* protein and EPI. With EPI treatments, the cell cycle of HepG2, A549, Panc02, and NIH-3T3 arrested in G2/M phase, and the distribution of both G0/G1 and S phases was reduced. Additionally, the cell cycle of HaCaT arrested in the sub-G1 phase and decreased in the G0/G1 phase. In 16HBE cells, the distribution of the cell cycle was arrested in S and G2/M phases, with a reduction in the G0/G1 phase (Figure 2). In contrast, *H. leucospilota* protein treatment only significantly affected the cell cycles of cancer cells (i.e., HepG2, A549 and Panc02) but had no evident effects on the three normal cells (i.e., NIH-3T3, HaCaT and 16HBE). For example, as shown in Figure 2, *H. leucospilota* protein treatment arrested the G0/G1 phase and reduced the proportion of the G2/M phase of HepG2 cells; the *H. leucospilota* protein treatment increased the G0/G1 phase and decreased the distribution of S and G2/M phases of A549 cells; the *H. leucospilota* protein treatment increased the S and G2/M phases and reduced G0/G1 phase of Panc02 cells.

### 3.3. Effects of H. leucospilota Protein Treatment on Inducing Cell Apoptosis and Nuclear Morphology Changes

From Figure 3, it is clear that the *H. leucospilota* protein treatment significantly induced the apoptosis of cancer cells (HepG2, A549, and Panc02), but had little effects on normal cells (NIH-3T3, HaCaT and 16HBE). Specifically, the apoptosis rate of all cancer cell lines investigated was about 40% when treated with *H. leucospilota* protein (Figure 3), whereas that of the normal cell lines was 7.16–29.65%, no significant difference compared with blank control. In contrast, EPI significantly induced the apoptosis and necrosis of all six cell lines, that of the rate was above 60% regardless of cell types (Figure 3).

Consistent with the results from the apoptosis assay, morphological analysis of the cancer cells after the treatment of *H. leucospilota* protein showed bright blue, nuclear fragmentation, and chromatin condensation, clear evidence suggesting apoptosis [28]; in contrast, morphological analysis did not show obvious morphological changes in any of the *H. leucospilota* protein treated normal cells. However, all six cell lines treated with EPI showed evident bright blue, nuclear fragmentation, chromatin condensation, and formation of apoptotic bodies (Figure 4).

### 3.4. Effects of H. leucospilota Protein Treatment on Cell Migration

As shown in Figure 5, both *H. leucospilota* protein and EPI treatments significantly inhibited cell migrations of all six cell lines (HepG2, A549, Panc02, NIH-3T3, HaCaT and 16HBE), and markedly reduced their proliferations. In addition, a closer analysis of the data suggested that the *H. leucospilota* protein treatment exhibited more pronounced inhibition in cell migrations (6.22% ± 1.85%–34.86% ± 3.15% (12 h) and 7.04% ± 1.70%–43.26% ± 7.18% (24 h)) than EPI (6.35% ± 2.69%–45.99% ± 5.68% (12 h) and 9.08% ± 2.73%–74.52% ± 7.91% (24 h)).

## 4. Discussion

Despite advances in molecular biology of cancers, improved diagnosis, and even optimal target therapies, the current treatment options for cancers are still insufficient to cure cancer patients. Conventional chemotherapy drugs, such as docetaxel, cabazitaxel, and epirubicin, and other widely used targeted therapy drugs, have led to concerns about drug resistance due to different mechanisms, such as p53 mutations [29] or deficiency [30], increased drug efflux by p-glycoprotein [31], enhancing DNA repair, and altering the signaling pathway of apoptosis [32]. As such, there are still urgent need for innovative anti-cancer drugs, especially in the unexplored areas of marine products.

Natural products, especially those derived from the marine environment, have recently and globally become well known in the last two decades for their pharmacodynamic potential in a variety of disease therapies such as cancer [33,34]. Several natural products have contributed significantly to the treatment of many cancers, including lung cancer, liver cancer, breast cancer, and averted multi-drug resistance [35,36,37]. Sea cucumber is one of the marine organisms belonging to the phylum Echinodermata; their bioactive compounds are attractive candidates for cancer chemoprevention and therapy; above all, the protein of sea cucumber is an effective source of bioactive peptides. It had been shown in previous studies show that peptides hydrolyzed from the body wall of the sea cucumber hold antioxidant, anti-inflammatory [18], and immunomodulating capacity [38] without exploring the anticancer effect. To our knowledge, the present study is the first to perform a comparative investigation into the effects of sea cucumber (*Holothuria leucospilota*) protein on the three cancer cell lines (HepG2, A549, Panc02) and the three normal cell lines (NIH-3T3, HaCaT and 16HBE) in cell viability, migration, cell cycle, cell apoptosis and cell morphology.

In the present study, *H. leucospilota* protein showed inhibition on the viability of the cancer cells in the manner of concentration or time- and concentration-dependency. Induction of apoptosis is one the most prominent markers of cytotoxic anti-tumor agents [39]. It has been shown that some natural compounds isolated from sea cucumber induce apoptotic pathways through several different mechanisms to inhibit cancer progression. Marine triterpene glycosides Stichoposide C induces apoptosis through the activation of Fas and caspase-8, cleavage of Bid, mitochondrial damage, and activation of caspase-3 [13]. Marzouqi et al. found that Frondoside A increased the sub-G1 (apoptotic) cell fraction through the activation of p53 and, subsequently, the caspases 9 and 3/7 cell death pathways in breast cancer cells [40]. We found that the *H. leucospilota* protein exerted its cytotoxic action through the induction of apoptosis, leading to the morphological changes in HepG2, A549, and Panc02 cancer cell lines. This process may be associated with the alteration of relevant pro-apoptotic factors, such as p53 [30], NF-ΚB [41], cleaved caspases-3, -8, and -9, cleaved PARP, and Bax [42]. However, the three normal cell lines (NIH-3T3, HaCaT and 16HBE) treated with *H. leucospilota* protein did not show induced cell apoptosis, suggesting that the targeting effect of *H. leucospilota* protein is to specifically induce tumor cells to undergo apoptosis while causing no damages to normal tissues. In contrast, all six cell lines treated with EPI underwent induced apoptosis and necrosis, suggesting that EPI possesses a toxic effect on both normal and cancer cells. Future studies will be necessary to explore possible mechanisms of *H. leucospilota* protein-induced apoptosis.

The *H. leucospilota* protein affects cell viability, altering the cell apoptosis and cell cycle distribution in cancer cells. Mammalian cells progress through several cell cycle phases (G1, S, G2, and metaphase) during cellular division. Cell cycle checkpoints, notable features of cell cycle progression, provide essential surveillance to prevent cells from entering the next phase before the previous phase has been completed [43,44]. Thus, halting the cell cycle can lead to prevention of cancer cell growth and division and is one of the major strategies for cancer therapy. It is likely that *H. leucospilota* protein may act as cell cycle non-specific drugs that inhibits cancer cell growth by targeting multiple phases of the cell cycle. Our results demonstrated that *H. leucospilota* protein could arrest the G2/M phase and G0/G1 phase in HepG2 cells, thereby promoting cell apoptosis and inhibiting tumor invasion and metastasis. It has also been demonstrated that *H. leucospilota* protein not only increased G0/G1 phase but also induced growth inhibition at S and G2/M phase with significant apoptosis in A549 cells. The cell cycle of Panc02 cells were arrested in S and G2/M phases, while the G0/G1 phase was reduced after being treated with the IC_50_ concentration of *H. leucospilota* protein. It is interesting to note that the *H. leucospilota* protein showed a greater selectivity and higher cytotoxicity than EPI towards cancer cells compared to normal cells.

It has been reported that inhibition of cell cycle kinases or regulation of the lever of cyclins can trigger tumor cell apoptosis, in addition to cell cycle arrest. The cell cycle regulators, such as cyclins or cyclin-dependent kinases, play an important regulatory role in cancer cell metabolism [45]. For example, Cyclin D1, an important regulator of cell cycle, is over-expressed with high frequency in many human cancers, targeting pathways such as the programmed cell death protein-1 (PD-1) axis [46]. The sulfated polysaccharide isolated from sea cucumber *Stichopus japonicus* on the 6-hydroxydopamine-induced apoptosis in SH-SY5Y cells (a human dopaminergic neuroblastoma cell line) significantly elevated protein levers of Cyclin D3 [47]. It has been reported that, in sea cucumber (*Apostichopus japonicus*), regulating Cyclin A could arrest cell cycle progression [48]. In HepG2, A549, and Panc02 cells, the effects of the distribution of cell cycle by *H. leucospilota* protein may be associated with several cyclins, cyclin-dependent kinases and other regulatory mechanisms. However, the results of cell cycle assay exhibited no significant effect of *H. leucospilota* protein on cell cycle distribution of normal cells (NIH-3T3, HaCaT and 16HBE (*p* > 0.05)). The specific mechanism needs further investigation in the future.

Based on our findings, *H. leucospilota* protein demonstrated anti-cancer properties through multiple mechanisms, including cytotoxic activity, induction of apoptosis, cell cycle arrest, suppression of invasion. Malignant cancer is highly aggressive and metastatic and continues to be an incurable leading cause of mortality [39]. Since tumor spreading is responsible for 90% of all human cancer deaths, development of therapeutic agents that inhibit tumor invasion or metastasis is highly desirable. We showed that the IC_50_ concentrations of *H. leucospilota* protein could impair cell migration in HepG2, A549, Panc02, NIH-3T3, HaCaT and 16HBE cells 12 and 24 h after treatment, and that the protein’s ability to inhibit cell migration was superior to EPI. We provided here evidence that *H. leucospilota* protein exerts a stronger inhibitory effect on the migratory properties of the six cell lines. It is likely that *H. leucospilota* protein could counter both the locally invasive and metastasizing behavior of tumor cells, which holds promising anti-metastatic potential for cancer therapy.

As is well known, traditional Chinese medicine (TCM) compounds have been used for a long time and are, traditionally, regarded as non-toxic and effective. Coincidentally, TCM exerts its effect through multiple targets and pathways with multiple components, rather than a single compound or herb. Sea cucumbers (*H. leucospilota*) have been extensively used in traditional Chinese medicine or as a dietary supplement; it has been reported that sea cucumber intestinal peptide induces apoptosis of MEC-7 breast cells via inhibition of the PI3K/AKT signal transduction pathway [49]. Furthermore, the anti-cancer effect of the sea cucumber intestinal peptide has been shown to be modulated through inhibiting the overexpression of EGFR, PI3K, AKT1, and CDK4 and through inhibiting the malignancy of lung cancer by regulating miR-378a-5p-targeted tumor suppressor candidate gene 2 (TUSC2) [50,51]. It has also been shown that sea cucumber-derived peptides could play a role in alleviating oxidative stress in cells [52]. These studies provide an experimental basis for the further development of sea cucumber as an anti-cancer nutritional supplement and a new direction to explore the mechanism of food-derived active peptides in anti-cancer applications. However, the exact molecular mechanism for this still remains unclear.

Our research showed that *H. leucospilota* protein plays an anti-cancer role in various human cancers, which may be associated with complexes (total protein) of the body wall to exert synergistic action, rather than with a single protein. However, further research is still required to identify and elucidate the active ingredients of *H. leucospilota* protein, to detail underlying molecular mechanisms of targeted cancer cells and to carry out pharmacological experiments in vivo. It should also be noted that an appropriate drug delivery method would also be required for the potential of the *H. leucospilota* protein as an anti-cancer treatment to be fully realized.

## 5. Conclusions

The marine-derived natural product sea cucumber (*Holothuria leucospilota*) protein showed potential anti-cancer activity toward HepG2, A549, and Panc02 cells and was more effective in reducing cell migration than EPI (the clinical anticancer drug). The targeted effects of *H. leucospilota* protein on cancer cells, including induction of apoptosis, suppression of metastasis on tumor cells, as well as the effect of cell cycle arrest make the *H. leucospilota* protein a promising candidate for the treatment and prevention of human cancers.

## Figures and Tables

**Figure 1 nutrients-14-00786-f001:**
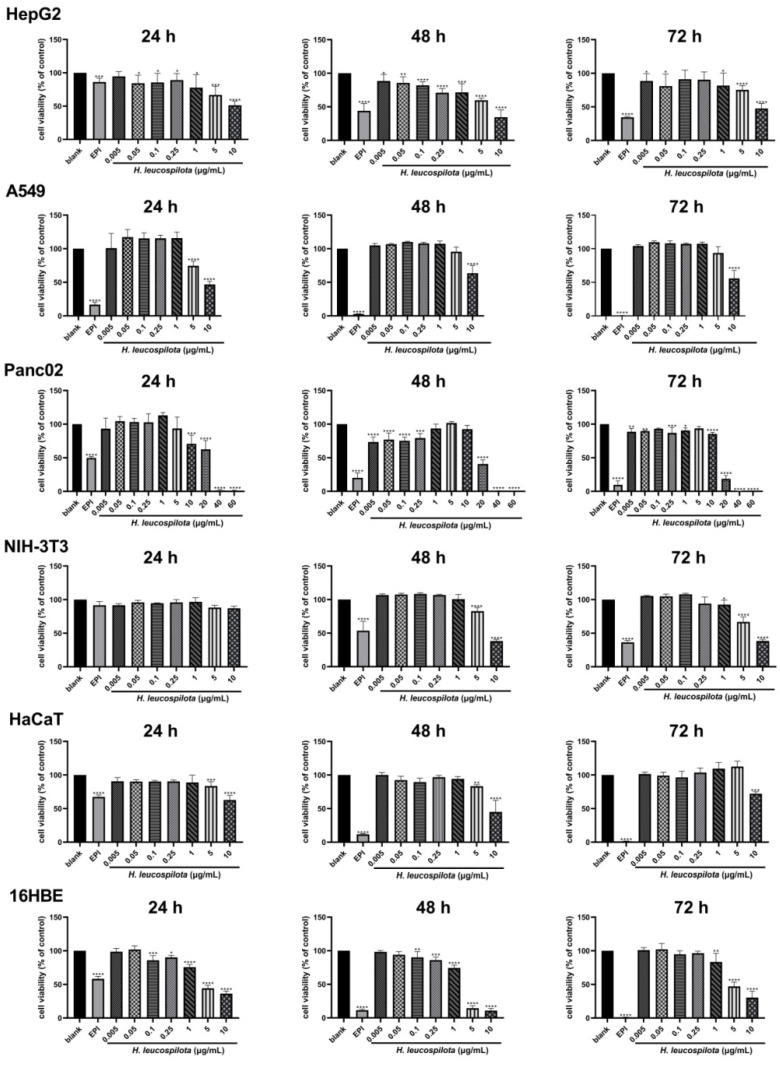
Inhibition of cellular viability by *Holothuria leucospilota* (*H. leucospilota*) protein. Exponentially growing HepG2, A549, Panc02, NIH-3T3, HaCaT and 16HBE cells were treated with EPI (10 μΜ) and the increased concentrations of *H. leucospilota* protein for 24, 48, 72 h. Viable cells were assayed as described in Materials and Methods. All experiments were repeated as least three times. Data are expressed as Mean ± SD. * *p* < 0.05, ** *p* < 0.01, *** *p* < 0.001, **** *p* < 0.0001.

**Figure 2 nutrients-14-00786-f002:**
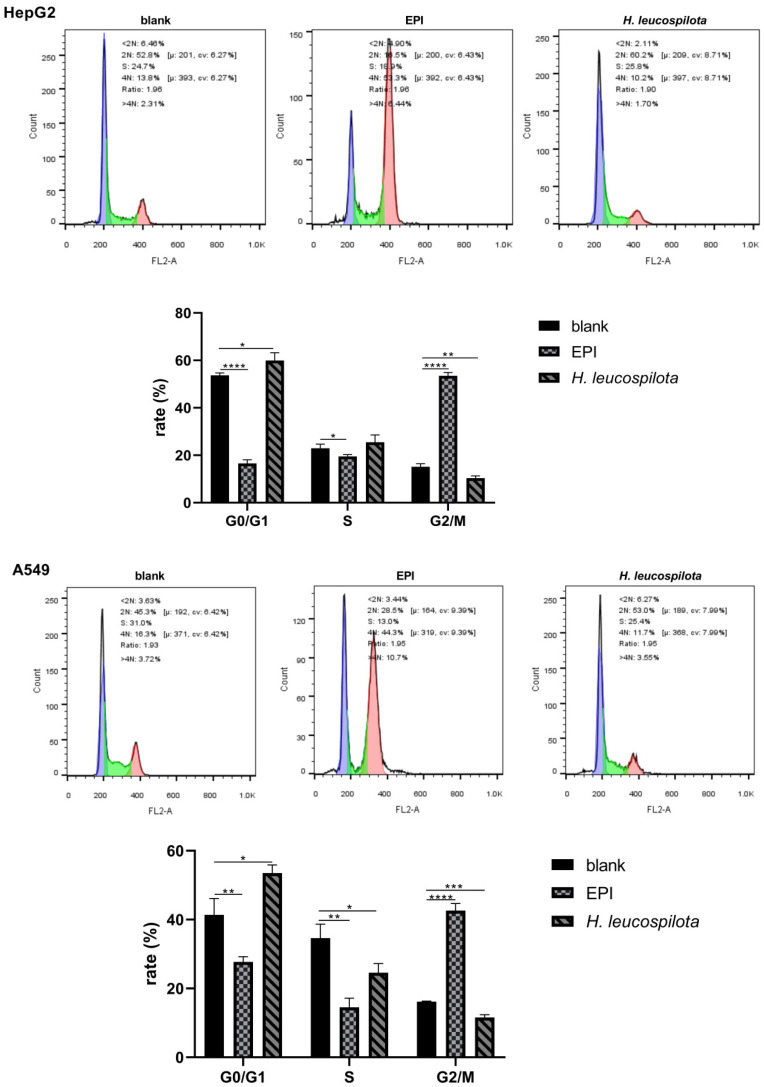
Effect of *H. leucospilota* protein on cell cycle distribution. Cells were treated with the IC_50_ concentrations of *H. leucospilota* protein for 48 h, with EPI (10 μΜ) as positive control. All experiments were repeated at least three times. Data are mean ± SD. * *p* < 0.05, ** *p* < 0.01, *** *p* < 0.001, **** *p* < 0.0001.

**Figure 3 nutrients-14-00786-f003:**
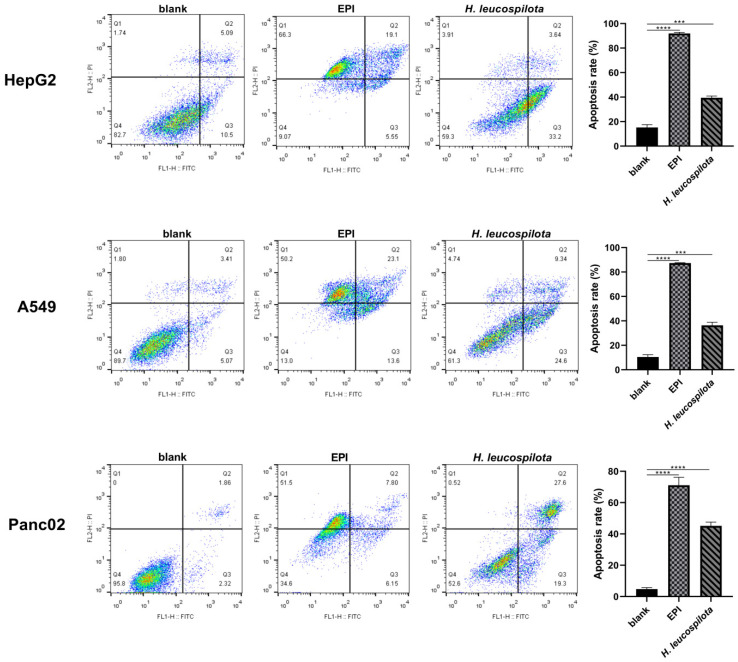
Effect of *H. leucospilota* protein treatment on cell apoptosis. Cells were treated with the IC_50_ concentrations of *H. leucospilota* protein for 48 h, and EPI (10 μΜ) as positive group. * *p* < 0.05, *** *p* < 0.001, **** *p* < 0.0001.

**Figure 4 nutrients-14-00786-f004:**
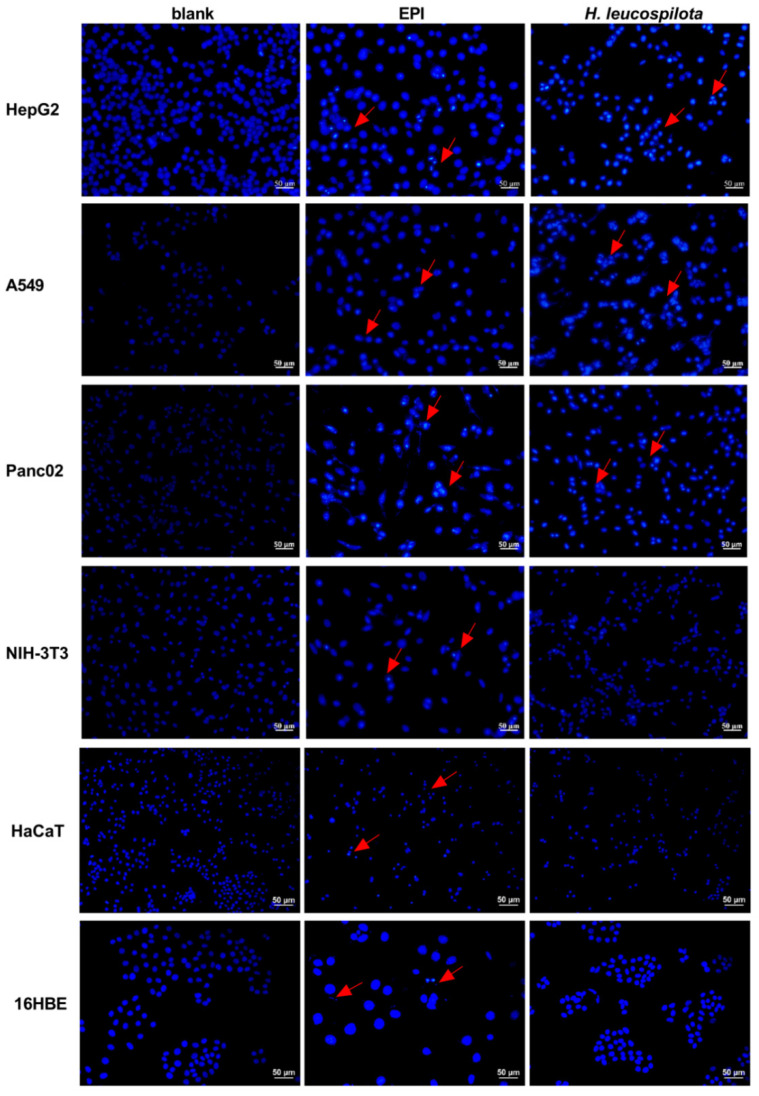
Effect of *H. leucospilota* protein treatment on cell morphology. Cells were treated with the IC_50_ concentrations of *H. leucospilota* protein for 48 h, and EPI (10 μΜ) as positive group. The morphological characteristics of apoptosis were shrunken, hyperchromatic, and pyknotic cells.

**Figure 5 nutrients-14-00786-f005:**
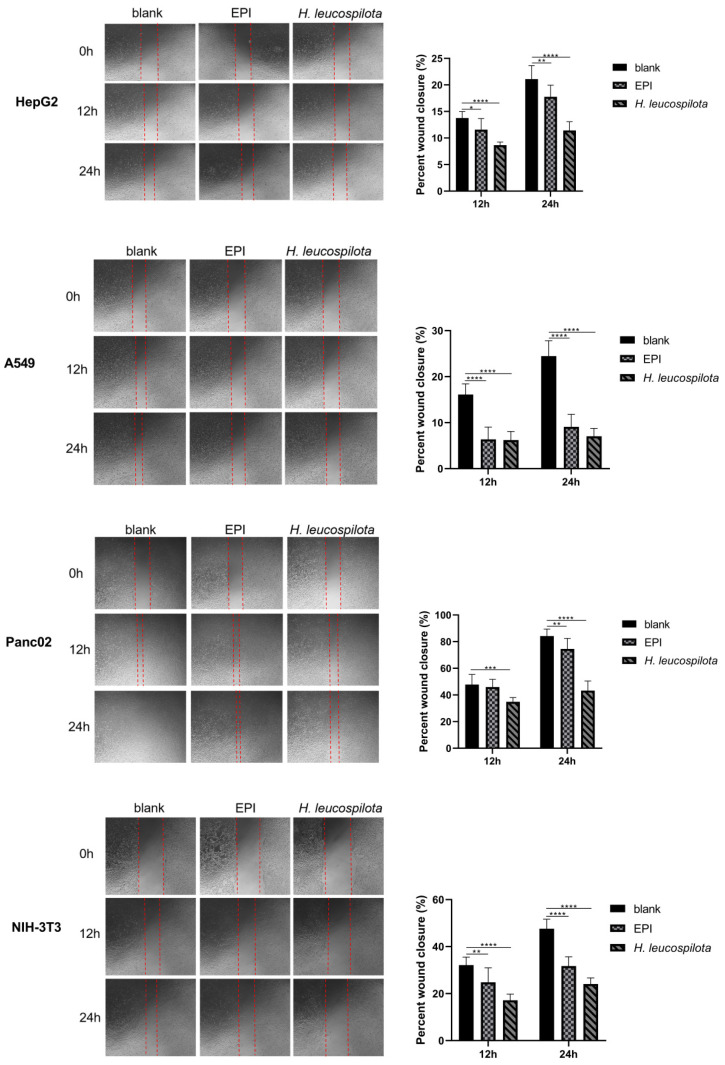
Effect of *H. leucospilota* protein treatment on cellular migration. Wounds were introduced in HepG2, A549, Panc02, NIH-3T3, HaCaT and 16HBE confluent mono-layers cultured in the presence or absence (control) with the IC_50_ concentrations of *H. leucospilota* protein for 0, 12, and 24 h, and EPI (10 μΜ) as positive group. Migration rate of HepG2, A549, Panc02, NIH-3T3, HaCaT, 16HBE in Figure. Experiments were repeated at least three times. Data are expressed as mean ± SD. * *p* < 0.05, ** *p* < 0.01, *** *p* < 0.001, **** *p* < 0.0001.

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
