# Peer review of "Cancer Cell Inhibiting Sea Cucumber (Holothuria leucospilota) Protein as a Novel Anti-Cancer Drug"

_nutrients, 2022, doi:10.3390/nu14040786_

Round 1

Reviewer 1 Report

Ru et al., describe the anti-cancer activity of sea cucumber proteins against different cancer lines. The authors perform several experiments including cell viability, flow cytometry and microscopy to validate the anti-cancer activity. While these experiments are relevant and important, the conclusions made by the authors are farfetched, and additional experiments / validation are required to demonstrate that H. leucospilota protein extract has higher anti-cancer activity in comparison to EPI.

  • Line 59 - The authors mention ‘Marine creature is enriched ore of drug development, and at least 10% of the marine 59 organism extracts have anti-tumor activity’. Authors need to provide relevant references to backup these claims
  • Line 64 – cytarabine is an FDA approved drug, but the authors incorrectly mention that it is under evaluation in clinical trials
  • Line 115 – the name of the media is incorrectly spelled as RPIM-1640, and should be corrected to RPMI-1640
  • Line 179 – ‘For example, for HepG2 cells, at all concentration studied (i.e. 0.005-10 μg/mL) H. leucospilota protein showed cytotoxicity similar to EPI at 10 μg/mL’ This statement is false. The cytotoxicity is similar to the control EPI only at 24 hours. At 48 and 72 hours, EPI is shows significantly higher toxicity
  • Line 183 – ‘In contrast, for the case of Panc02 cells, H. leucospilota protein demonstrated significant cyctotoxicity even at low concentrations (i.e. 0.005-1 μg/mL) after 48 and 72 h (P <0.001), effects similar to EPI at concentrations above 20 μg/mL at 24, 48, 72 h.’ This statement is false. The cytotoxicity of EPI is significantly higher at 48 and 72 h
  • Figure 4 – the cell sizes for the same cell lines across blank, EPI and H. leucospilota look very different. The authors need to size the images correctly before analyzing the results
  • Figure 5 – the images are unclear and is hard to understand the results
  • Line 244 – ‘In addition, a closer analysis of the data suggested that the H. leucospilota protein treatment exhibited more pronounced inhibition in cell migrations (6.22% - 34.86% (12 h) and 7.04% - 43.26% (24 h)) than EPI (6.35% 246 - 45.99% (12 h) and 9.08% - 74.52% (24 h))’ Since the authors have not performed head to head statistical analysis of EPI with H. leucospilota, these claims can not be made
  • In addition to these experiments, the authors should also perform a western blot to determine levels of apoptotic markers such as Bcl-2, Bid, Bax, caspase, PARP and others to validate the apoptotic mechanism of H. leucospilota extract
  • The authors should cite recent publications on bioactive natural products such as:
    • Sharma S et al., Anti-Cancer Agents in Medicinal Chemistry, 2022. 21(3); 288-315
    • Singh et al., Bioorg Med Chem, 2019. 27(16): 3477–3510

Reviewer 2 Report

In the research paper entitled “Cancer Cell Inhibiting Sea Cucumber (Holothuria Leucospilota) Protein as A Novel Anti-Cancer Drug”, the authors tried to show the anticancer effect of protein extract from Holothuria Leucospilota. There are a few points to be noted:

  1. The authors should answer why they used the protein extract and not some secondary metabolite. Proteins act as immunogens and their clinical application is very limited due to the raise of antigen-antibody interaction.
  2. In materials and methods section 2.6: it is FACS and not FASC. We can call annexin staining as flow cytometry and usage of FACS is inappropriate as there is no sorting involved in the technique.
  3. The protein extract did not show dramatic alteration in the cell cycle even with high concentration.
  4. The migration assay graphs were not clear.

Round 2

Reviewer 2 Report

The authors tried to improve the suggestions. The authors highlighted the short bioactive peptides in the protein mixture. The main cons of the paper are the lack of molecular mechanism and identification of the peptides. Usually, when we mention superfood, it is rich in bioactive molecules and authors tried to show the sea cucumber as a super food. But the problem is that the ingestion of food causes a lot of breakdowns and we don't know in what form, the molecules enter into the bloodstream, reach cancer cells and exert their anticancer activity. 

Author Response

We thank the reviewer for the comments. It is a very important point.

We realize that this is an important point. To address this point, additional text has been added in the discussion to indicate the importance of an appropriate method of drug administration, in Lines 381 to 383.